# Dietary Acrylamide Intake Alters Gut Microbiota in Mice and Increases Its Susceptibility to *Salmonella* Typhimurium Infection

**DOI:** 10.3390/foods10122990

**Published:** 2021-12-03

**Authors:** Zhen Wang, Hongxu Liu, Jiaxiu Liu, Xiaomeng Ren, Guoku Song, Xiaodong Xia, Ningbo Qin

**Affiliations:** 1Collaborative Innovation Center of Seafood Deep Processing, National Engineering Research Center of Seafood, School of Food Science and Technology, Dalian Polytechnic University, Dalian 116304, China; 18854802285@163.com (Z.W.); liuliuhongxu@163.com (H.L.); liujx94@163.com (J.L.); rxmfighting@163.com (X.R.); songguokuxx@163.com (G.S.); xdxia@nwsuaf.edu.cn (X.X.); 2Sino-US Joint Research Center for Food Safety, College of Food Science and Engineering, Northwest A & F University, Xianyang 712100, China

**Keywords:** acrylamide, *Salmonella* Typhimurium, infection, gut microbiota, inflammatory cytokines, tight junction proteins

## Abstract

Acrylamide (AA) has been extensively examined for its potential toxicological effects on humans and animals, but its impacts on gut microbiota and effects on hosts’ susceptibility to enteric infection remain elusive. The present study was designed to evaluate the effect of AA on gut microbiota of mice and susceptibility of mice to *S*. Typhimurium infection. After four weeks’ intervention, mice fed with AA exhibited significantly decreased body weight. Meanwhile, 16S rRNA gene sequencing showed reduced relative abundance of Firmicutes and increased abundance of Bacteroidetes in AA-treated mice prior to infection. In addition, we observed high relative abundance of *Burkholderiales* and *Erysipelotrichales*, more specifically the genus *Sutterella* and *Allobaculum*, respectively, in AA-treated mice before infection. Subsequently, the mice were orally infected with *S*. Typhimurium. The histological changes, systemic dissemination of *S*. Typhimurium, and inflammatory responses were examined. Compared to mice fed with normal diet, mice fed AA exhibited higher level of bacterial counts in liver, spleen, and ileum, which was consistent with exacerbated tissue damage determined by histological analyses. In addition, higher expression of pro-inflammaroty cytokines, p-IκBα, and p-P65 and lower mRNA expressions of *mucin2*, *occludin*, *zo-1*, *claudin-1*, and *E-cadherin* were detected in AA-treated mice. These findings provide novel insights into the potential health impact of AA consumption and the detailed mechanism for its effect on *S*. Typhimurium infection merit further exploration.

## 1. Introduction

The mature, complex gut microbiota inhabiting the mammalian intestinal tract has an important role on the host immune system and defense functions during infection [1]. These gut microbiota has a large impact on the prevention of pathogen infection by direct and indirect mechanisms [2]. These microbiota-pathogen interactions include production of inhibitory metabolites and antagonism or competition for nutrient niche and space, but also competition of physical niches and glycosylation of mucus [3]. Dysbiosis of gut microbiota caused by environmental factors such as dietary changes could alter the composition and structure of gut microbiota and the amount of some metabolites to produce an effect on host susceptibility towards enteric pathogens infection [4,5,6].

*Salmonella* Typhimurium (*S*. Typhimurium) is a member of the family of gram-negative *Enterobacteriaceae*. Importantly, *S*. Typhimurium can cause systemic infection in mice and foodborne gastroenteritis in humans [7]. It has been widely used in a laboratory model to understand the biological mechanisms involved in multiple intracellular behaviors including invasion, proliferation, dissemination, and transmission to host cell and resultant host immune responses [8]. Previous studies have suggested that changes in composition and structure of gut microbiota have a significant impact on the intestinal infection with *S*. Typhimurium. For instance, some commensal microbiota in gastrointestinal produces colonization resistance against *S*. Typhimurium infection owning to antibiotic treatment [9,10]. Jacobson et al. indicated that composition of the gut microbiota alters intestinal expansion of *S*. Typhimurium and *Bacteroides* spp. mediate colonization resistance to *S*. Typhimurium infection [5]. The pathogenic potential of the pathogen, such as invasion and colonization, are likely to be influenced by many environmental factors including endogenous and exogenous physic-chemical signals in gastrointestinal phase.

Acrylamide (AA) had been recognized as potential carcinogen in 1994, which could cause lots of negative effects in human or animal organs such as brain, lungs, liver, skin and glands [11,12,13,14]. In recent decades, increased consumption of some food products by frying, roasting, and baking, with high caloric and AA intake, has been correlated with current serious health problem [15,16,17]. It is worth noting that fried potatoes, bakery products, and coffee were major dietary exposure to AA [18]. A diet rich in AA can increase the risk of several types of cancer and induce nephrotoxicity, oxidative stress, inflammation, apoptosis, and DNA damage [19,20,21]. However, whether AA intake influences the gut microbiota and susceptibility to enteric infection has received little attention.

Therefore, this study was designed to evaluate the effect of AA on gut microbiota and *S.* Typhimurium infection in mice. The body weight of mice was determined and compared between the normal control (NC) and AA groups. The contribution of AA on abundance and composition of gut microbiota was investigated by high throughput16S rRNA gene sequencing prior to infection. Then histological analysis was evaluated after oral *S.* Typhimurium infection. Moreover, the expression of genes encoding inflammatory markers (IL-6, IL-1β, and TNF-α), Mucin2 (Muc2), E-cadherin (E-cadh), and tight junction proteins including occludin (Ocln), zonula-occludens (ZO)-1, claudin-1 (Cldn-1) were analyzed by real-time quantitative polymerase chain reaction (RT-qPCR). In addition, the protein expression of p-IκBα, p-P65, and occludin were determined by immunoblot analysis.

## 2. Materials and Methods

### 2.1. Reagents

Acrylamide was purchased from Sigma-Aldrich (St. Louis, MO, USA). Mice were purchased from Liaoning changsheng Biotechnology Co., Ltd. (Shenyang, China) Luria-Bertani (LB) medium was purchased from Qingdao Hope Bio-Technology Co., Ltd. RNAeasy Animal Long RNA Isolation Kit was purchased from Beyotime Biotechnology Co., Ltd. (Shanghai, China). PrimeScript RT reagent Kit and TB Green Premix Ex TaqII Kit were purchased from Takara Co., Ltd. (Kusatsu, Shiga, Japan). PowerMax DNA extraction kit was purchased from MoBio Laboratories Inc. (Carlsbad, CA, USA). PicoGreen dsDNA assay kit was purchased from Thermo Fisher Scientific Inc. (Waltham, MA, USA). Antibodies to β-actin, p-NF-κB P65, and p-IκBα were purchased from Cell Signaling Technology (Beverly, MA, USA). Antibody to occludin was pruchased from Proteintech Group, Inc. (Rosemont, PA, USA).

### 2.2. Animals Treatment

All animal experiments were approved by the Ethics Committee of Animal Experimentation in Dalian Polytechnical University (Approval number: DLPU2020033). Four-weeks old C57BL/6J male mice were purchased from Liaoning Changsheng Biotechnology Co., Ltd. (Shenyang, China) (Certificate number: SCXK-2015-0001). The mice were allowed to receive food and water *ad libitum*, and maintained on a 12 h-light-dark cycle for 2 weeks with a temperature (24 ± 1 °C) and humidity (60 ± 5%). After acclimation, the mice were randomly divided into NC group with standard chow diet (28.6% of energy from protein, 13.4% of energy from fat, and 58.0% of energy from carbohydrates) and AA group with chow diet and oral AA (*n* = 8 pre group, four mice per cage). AA dissolved with sterile water was administered by oral gavage at a dose of 1 µg/kg per day, whilst same volume of sterile water was administered in NC group.

### 2.3. S. Typhimurium Infection

After four weeks, the mice were divided into four groups including NC, AA, NC-ST, and AA-ST group. NC-ST and AA-ST groups were infected by oral inoculation. Overnight culture of *S.* Typhimurium (SL1344) from our own laboratory collection was centrifuged at 8000× *g* for 3 min, washed twice with sterile water, and resuspended in sterile water. A 200 µL of *S.* Typhimurium (5 × 10^8^ CFU) was administrated once by a gavage needle. The mice were then monitored for five days.

### 2.4. Faecal Samples and Microbiota Profiling

Fecal samples were collected before infection in sterile tubes, snap frozen in liquid nitrogen, and transferred to −80 °C for 16S rRNA analysis. DNA was extracted from fecal samples with PowerMax DNA extraction kit. The concentration of DNA was determined with Nanodrop ND-1000 spectrophotometer (Thermo Fisher Scientific, Waltham, MA, USA). The variable V3/4 region of 16 S rRNA gene was amplified with following primers 319F (5′-ACTCCTACGGGAGGCAGCAG-3′) and 806R (5′-GGACTACHVGGGTWTCTAAT-3′). The PCR products were purified with AMPure XP beads (Beckman Coulter, Indianapolis, IN), quantified by the PicoGreen dsDNA assay kit, and sequenced using Illumina HiSeq 4000 with 2 × 300 bp paired-end reads as previous reported method [22]. For operational taxonomic units (OTUs), nucleotide sequence with 97% identity was assigned to the same OTUs based on SILVA128 database. Alpha diversity analysis was performed to assess the species diversity of different samples by four indices including ACE, Chao 1, Shannon, and Simpson, which were evaluated by the QIIME software. Beta diversity was analyzed with principal coordinate analysis (PCoA) plots based on the weighted UniFrac distance measurements. In addition, linear discriminant analysis effect size (LEfSe) analysis was carried out to investigate significant biomarkers.

### 2.5. Histological Analyses

Five days after infection, the mice were sacrificed with 300 μL of urethane. The tissues including liver, spleen, and ileum were collected and divided into two parts. One part was quickly freezed in liquid nitrogen and stored at −80 °C for following analysis. Another part was fixed with 4% paraformaldehyde for 24 h, dehydrated with different concentration of ethanol, and embedded in paraffin. For histological analysis, paraffin-embedded tissue sections were sectioned into 5 µm and stained with haematoxylin and eosin (H&E) according to standard procedures. Histological changes were analyzed and compared with a light microscope at 100× magnification. The histological score was adapted from previous reports with some modifications. The inflammation scores of the livers and spleens were as follows: 0 = normal, 1 = mild inflammation, 2 = moderate inflammation and 3 = marked inflammation. The ileal tissues were scored on a scale of 0–3 for two parameters to yield a maximum score of 6: infiltrate of inflammatory cells to the villi and crypt. The gradient of the inflammatory cell infiltration was used: 0 = normal, 1 = mildly or slightly increased, 2 = moderately increased and 3 = marked or highly increased [23].

### 2.6. RNA Extraction and RT-qPCR Analysis

Total RNA of samples from liver, spleen, and ileum were extracted with RNAeasy Animal Long RNA Isolation Kit according to the manufacturer’s protocol. RNA concentrations were determined by using a Nanodrop ND-1000 spectrophotometer (Thermo Fisher Scientific Inc., Waltham, MA, USA). Then, cDNA was synthesized by a PrimeScript RT reagent Kit according to the manufacturer’s protocol. Primer sequences were outlined in Appendix A. RT-qPCR reactions were performed in a 20 µL reaction system using TB Green Premix Ex TaqII. The thermal cycler parameters were 95 °C for 30 s, 45 cycles of 95 °C for 15 s, and 60 °C for 15 s.

### 2.7. Bacterial CFU Determination

After mice were infected for five days, mice were euthanized and some tissues including liver, spleen, and ileum were removed and homogenized in stomacher bags at 1 mg per mL LB, and 100 µL was plated on LB plates pre-treated with streptomycin for *S.* Typhimurium CFU determination. After incubated in 37 °C for 24 h, bacterial colonies were counted.

### 2.8. Western Blot Analysis

The ileum tissue sample was weighed, homogenated, and lysed in RIPA buffer with 1% PMSF and 0.8% phosphatase inhibitors. The homogenate was centrifuged at 10,000 rpm for 5 min at 4 °C. The supernatant was collected after centrifugation. The concentrations of the proteins were determined by BCA protein assay kit (Solarbio, Beijing, China). The proteins were separated by 10% SDS-PAGE and transferred to PVDF membranes, which were blocked with 5% skimmed milk solution for 1 h at room temperature. The membranes were incubated overnight with primary antibody at 4 °C, washed with TBST for 4 times, and incubated with secondary antibody for 1 h. The enhanced chemiluminescence system (Beyotime, Shanghai, China) was used to visualize the protein bands.

### 2.9. Statistic Analysis

Statistical difference was analyzed using GraphPad Prism 7.0. Data were showed with means ± SEM. One-way ANOVA was conducted with Sidak’s test for multiple comparisons. Two-way ANOVA was conducted with Bonferroni test for multiple comparisons. The comparison of 16S rRNA was performed by the Mann-Whitney and Kruskal-Wallis test. Significance was shown as * *p* < 0.05, ** *p* < 0.01, and *** *p* < 0.001.

## 3. Results

### 3.1. The Effect of AA on the Body Weight, Food and Water Intake

To test the effect of AA on body weight, we designed a study where C57BL/6J mice were fed either a normal chow or chow diet with oral AA for a relatively short period of 4 weeks in order to avoid alterations in peripheral and central nervous systems associated with long-term feeding in the model [24,25]. Compared with the NC group, mice fed with AA showed a significant decreased body weight (*p* < 0.001) (Figure 1A) while there was no significant difference in food and water intake between NC and AA groups (Figure 1B,C).

### 3.2. The Effect of AA on the Diversity and Composition of Gut Microbiota

The composition and abundance of gut microbiota of mice was analyzed and compared by 16S rRNA amplicon sequencing analyses in NC and AA groups. A total of 962 operational taxonomic units were identified in 13 fecal samples (Figure 2A). Sequencing analysis of fecal samples from AA group mice produced an average of 255.333 ± 11.885 observed species compared to NC group (375.500 ± 6.985) (Appendix A). The community richness (Chao1 *p* = 0.0221 and Ace index *p* = 0.0232) and diversity (Shannon *p* = 0.6282 and Simpson index *p* = 0.7308) were assessed by α-diversity analysis (Figure 2B–E). Abovementioned results indicated that AA group showed no significant effect on the gut microbiota diversity compared to NC group. The β-diversity analysis based on the principal coordinate analysis (PCoA) of the weighted UniFrac distance showed significant difference between NC and AA group (Figure 2F). LefSe analysis was used to note distinct biomarkers and reveal significant microbiome differences between the NC and AA group (Figure 3A). As shown in Figure 3B, we observed that the relative abundance of Burkholderiales, Betaproteobacteria, Sutterella, Bifidobacterium, Bifidobacteriales, Bifidobacteriaceae, Actinobacteria, Erysipelotrichales, Erysipelotrichi, and Allobaculum was higher in AA group but the relative abundance of Enterobacteriaceae and Enterobacteriales was higher in NC group.

At the phylum level, there was no significant difference between NC and AA groups in relative abundance of Bacteroidetes, Firmicutes, Proteobacteria, Actinobacteria, and Deferribacteres except Verrucomicrobia, which showed 55-fold higher relative abundance in AA group (Figure 3C,D and Appendix A). At the class level, AA-treated mice showed 78.8-, 21.3-, 14.8-, and 55-fold higher relative abundance of Actinobacteria, Erysipelotrichi, Betaproteobacteria, and Verrucomicrobiae (Appendix A). At the order level, AA-treated mice showed 211.8-, 21.3-, 14.8-, and 55-fold higher relative abundance of Bifidobacteriales, Erysipelotrichales, Burkholderiales, and Verrucomicrobiales (Appendix A). At the family level, AA-treated mice showed 211.8-, 21.3-, 15.3-, and 55-fold higher relative abundance of Bifidobacteriaceae, Erysipelotrichaceae, Alcaligenaceae, and Verrucomicrobiaceae and 3.4- and 3.0-fold lower abundance of Bacteroidaceae and Rikenellaceae (Appendix A). At the genus level, AA-treated mice showed 211.8-, 373.6-, 15.9-, and 54.9-fold higher relative abundance of Bifidobacterium, Allobaculum, Sutterella, and Akkermansia and 3.4-fold lower abundance of Bacteroides (Figure 3E and Appendix A).

### 3.3. The Effect of AA on Host Susceptibility to S. Typhimurium Infection

We subsequently investigated the host susceptibility to oral *S.* Typhimurium infection after AA consumption. As shown in Figure 4, the level of *S.* Typhimurium in some internal organs such as liver (*p* < 0.001), spleen (*p* < 0.001), and ileum (*p* < 0.01) were significantly increased in mice with AA diet compared to that of NC group. Moreover, the level of *S.* Typhimurium in liver and spleen was significantly higher than that in ileum in both NC-ST and AA-ST group. The results indicated that feeding of AA diet for 4 weeks significantly increased host susceptibility to oral *S.* Typhimurium infection compared to NC group.

To further investigate the potential links between diet change and the host response to oral *S.* Typhimurium infection, we measured histological changes in internal organs of host after infection. Histological studies demonstrated that hepatocytes were morphologically incomplete and nuclei were partly enlarged and leaky in liver of AA-ST group compared to NC-ST group. In addition, the hepatic portal vein and hepatic sinuses were markedly dilated and filled with a large number of red blood cells and inflammatory cells in liver of AA-ST-treated mice (Figure 5A). Moreover, there was significant difference on histology score between AA and AA-ST group (Figure 5B). There was no significant difference on histology score between NC and AA/NC-ST group, but there was mild hepatic multifocal necrosis in AA group (Figure 5A). We observed complete splenic corpuscles, red and white pulps, and little inflammatory cells in NC-ST group. However, there were no complete splenic corpuscles and clear boundary between the red and white pulps in AA-ST group. Furthermore, a large number of inflammatory cells were infiltrated in spleen in AA-ST group (Figure 5A). There was no significant difference on histology score between NC and AA/NC-ST group (Figure 5C). In ileum of normal infected mice, we observed complete villi, crypt, mucosa, lots of goblet cells, and little of inflammatory cells. However, the villus was significantly destroyed and amounts of inflammatory cells were infiltrated in mucosa in ileum of AA-ST group (Figure 5A). There was no significant difference between NC and NC-ST group, but there was significant difference on histology score between NC and AA group (Figure 5D). These results indicated that AA may aggravate pathological changes in oral *S.* Typhimurium infection mice.

### 3.4. The Effect of AA on Inflammatory Cytokines

To evaluate the gene expression profile of some target genes, we compared the gene expression level of NC, AA, NC-ST, and AA-ST group. As shown in Figure 6A–C, expression level of genes encoding pro-inflammatory cytokines such as IL-6, TNF-α, and IL-1β showed no significant difference in liver, spleen and ileum between NC and AA group. However, expression level of pro-inflammatory cytokines showed significant difference in liver, spleen and ileum between AA and AA-ST group. This is consistent with histological analysis of inter organs, where increased inflammatory cells were infiltrated in responding to oral *S.* Typhimurium infection in mice fed with AA. The NF-κB signaling pathway is important in inflammation. Under the activation, the IκBα proteins were phosphorylated and then degraded to lead to the translocation of phosphorylated P65 from cytoplasm to nucleus and release of inflammatory cytokines. Thus, we compared the expression level of NF-κB signaling pathway-related proteins between NC-ST and AA-ST group. The results indicated the phosphorylated protein levels of IκBα and P65 were significantly higher in AA-ST group than that of NC-ST group (Figure 7).

### 3.5. The Effect of AA on Intestine Epithelial Structure

The release of inflammatory cytokines may be related with the damage of intestine epithelial structure. Therefore, we examined the transcriptional expression of Muc2, E-cadherin, and tight junction proteins such as occludin, ZO-1, claudin-1 in ileum. The mRNA abundance of Muc2, E-cadherin, ZO-1, occludin, and claudin-1 showed no significant change in ileum between NC and AA groups. However, the mRNA abundance of Muc2 (*p* < 0.001), E-cadherin (*p* < 0.001), ZO-1 (*p* < 0.001), occludin (*p* < 0.001), claudin-1 (*p* < 0.001) was significantly reduced in ileum in AA-ST group compared to that of AA group (Figure 8). In addition, the expression of occludin was remarkably suppressed in AA-ST group compared to that of NC-ST group (Figure 7).

## 4. Discussion

Generally, in food manufacturing, thermal processes are often used to maintain a prolonged shelf-life and make a good effect on the final quality of foods. However, some heating processes such as baking, frying, and roasting often resulted in the formation of undesired compounds. These compounds are not naturally present in foods, but that may develop during heating processes with different chemical reactions and cause harmful effects such as mutagenic and carcinogenic effects. One of the most well-known compounds is AA, which was formed by Maillard reaction at high temperatures in carbohydrate-rich foods such as French fries, potato chips, bread, and coffee, etc. [11]. AA, as a neurotoxin and potential carcinogen, has been found to increase the risk for many types of disease in rodent models [26]. However, the exposure of dietary AA has not been associated with pathogen infection and actual risk of AA for pathogen infection is lack of convincing evidence.

In present study, C57BL/6J mice were fed either a normal diet with AA or regular chow for 4 weeks prior to infection. Indeed, the body weight of mice was different in two groups. The body weight of mice fed with AA was significantly decreased during experimental period compared to that of NC group. Previous evidence had demonstrated that Firmicutes/Bacteroidetes ratio was remarkably increased in obese hosts but decreased in lean hosts [27,28,29]. We observed that AA induced a decrease in the relative abundance of Firmicutes but increase in the relative abundance of Bacteroidetes compared with that of NC group. Moreover, the F/B ratio was significantly decreased in AA-treated group. Therefore, AA-induced weight loss may be correlated with the abundance and composition of gut microbiota.

Subsequently, the mice were orally infected with *S*. Typhimurium. *S*. Typhimurium, as a model pathogen, engages in a complex fashion with hosts to cause disease [30]. The mouse is a natural host of *S.* Typhimurium, which causes a systemic infection in susceptible mice that is similar to typhoid fever caused by *S.* Typhi in humans [31]. Generally speaking, oral infection with *S.* Typhimurium in susceptible mice may lead to systemic infection though the associated lymphoid tissues. However, in normal mouse models, *S*. Typhimurium may not efficiently colonize in the ileum, which was caused by colonization resistance of commensal microflora of intestinal tract in mice [30,32]. Afterwards, several studies indicated that antibiotic especially streptomycin pretreated mice may develop good infectious model in respond to oral *S.* Typhimurium. In present study, we observed that the level of *S.* Typhimurium is less than 10^5^ CFU/g colonized in ileum, which suggested that commensal microflora may disrupt and resist colonization of *S.* Typhimurium in normal mice. However, a higher *S.* Typhimurium burden was found in the liver, spleen, and ileum of mice in AA-ST group, which indicated that AA could promote the colonization of *S.* Typhimurium to cause systematic infection. Of course, these phenomena were confirmed by less tissue destruction and histological changes in liver, spleen, and ileum of mice in NC-ST and AA-ST group. Moreover, a diffuse inflammation associated with villi damage and inflammatory infiltrate had been observed in systematic histopathological analyses of AA-pretreated mice with oral *S.* Typhimurium infection. As shown in Figure 3, we observed that the relative abundance of the class *Betaproteobacteria*, the order *Burkholderiales*, and the genus *Sutterella* were significantly higher in AA group compared with NC group. The order *Burkholderiales* is phenotypically, metabolically, and ecologically diverse and plant, animal, and human pathogens, containing the families *Burkholderiaceae*, *Oxalobacteraceae*, *Alcaligenaceae*, *Sutterellaceae*, and *Comamonadaceae* [33]. Among them, various genera and species occasionally caused serious invasive diseases like septicemia and pneumonia in susceptible individuals [34]. Bacteria of the genus *Sutterella* had been identified in human feces, which had been associated with some gastrointestinal disturbance and infections [35,36,37]. In addition, the relative abundance of the class *Erysipelotrichi*, the order *Erysipelotrichales*, the family *Erysipelotrichaceae*, and the genus *Allobaculum* were higher in AA-treated mice than that of NC group. Erysipelotrichaceae plays an important role in both human and animal studies related to metabolic disorders [38,39,40,41]. Previous studies showed that the abundance of Erysipelotrichaceae was significantly increased in C57BL/6J mice infected with *Toxoplasma gondii* or *Giardia muris* or TNF-divern Crohn’s disease (CD)-like transmural inflammation [42,43]. Therefore, these findings are of interest and show that AA might promote the colonization and infection of *S.* Typhimurium by affecting the abundance and composition of gut microbiota.

It was well known that the infiltration of pro-inflammatory cytokines often exacerbated inflammation and resulted in tissue damage in mice with *S.* Typhimurium infection. The up-regulation of pro-inflammatory cytokine activity may exacerbate tissue impairment caused by *S.* Typhimurium [44]. In present study, we observed that the expression level of pro-inflammatory cytokines in the liver, spleen, and ileum was significantly increased in *S.* Typhimurium infected mice treated with AA. In addition, we compared the phosphorylated protein levels of IκBα and P65 between AA-treated and normal mice after *S.* Typhimurium infection. These data demonstrated that the exacerbated *S.* Typhimurium infection caused by AA may be associated with enhanced expression of p-IκBα and p-P65 and their downstream pro-inflammatory cytokines.

The intestinal epithelial and endothelia cells could form physical barriers that are essential to intestinal homeostasis and protect organisms from external environment and harmful luminal products [45]. The barrier function requires the formation of tight junctions (TJs) and adherens junctions. TJs are necessary for establishing a barrier function in gut as it can regulate paracellular passage of molecules and restrict the diffusion of some plasma membrane proteins and lipids to main homoeostasis in organs and tissues [46]. TJs are highly dynamic structures and thus it is not surprising that perturbation of TJ proteins expression is associated with a lot of diseases [45,47,48,49]. Additionally, it is known that TJ components were targeted by various pathogenic bacteria, which enter and infect cells by reorganization or degradation of related TJ proteins. Previous studies have demonstrated that TJs are composed of transmembrane proteins including occludin, claudins, junctional adhesion molecule 1(JAM-1), cytoplasmic plaque proteins including zonula-occludens (ZO)-1, -2 and -3, and some several signaling proteins [50,51]. In present study, we observed that the mRNA expression of occludin, ZO-1, and claudin-1 and protein expression of occludin was significantly lower in AA-ST group than that in AA group, which indicated that AA was helpful for infection/invasion of *S.* Typhimurium to TJ proteins in ileum. Cell–cell adhesion is mostly composed of tight junction and adherens junctions. Epithelial cells are characterized by strong cell-cell adhesion [52]. On the basis of previous studies, we found that E-cadherin is very important for the assembly of special cell-cell junctions [53]. The invasion of pathogen is achieved via heterophilic bind between bacterial internalin protein and E-cadherin of host [54]. The mRNA expression level of E-cadherin was reduced in AA-ST group in the experiment. Mucins, located on the surface of the intestinal epithelium, play a critical role in maintaining physical barrier function of epithelial cells and protecting layer against pathogen translocation and invasion. Muc2 was known as an important regulatory factor in gut immune system and gene expression had been widely studied [55]. Therefore, we investigated the role of Muc2 in host defense against *S.* Typhimurium infection. Indeed, our results showed the Muc2 gene expression was significantly down-regulated in AA-ST group compared to that in AA group. Aforementioned results are positively correlated with severe damage and higher level of bacterial counts in tissues and indicate that AA might exacerbate the *S.* Typhimurium infection.

## 5. Conclusions

In conclusion, consumption of AA could increase susceptibility of mice to *S.* Typhimurium infection, and these effects are possibly associated with altered gut microbiota composition, enhanced expression of pro-inflammatory cytokines, p-IκBα, and p-P65, and decreased expression of tight junction proteins. Uncovering the detailed mechanism in future research may extend our understanding of health repercussions of AA and may provide new avenues for preventing *Salmonella* infection.

## Figures and Tables

**Figure 1 foods-10-02990-f001:**
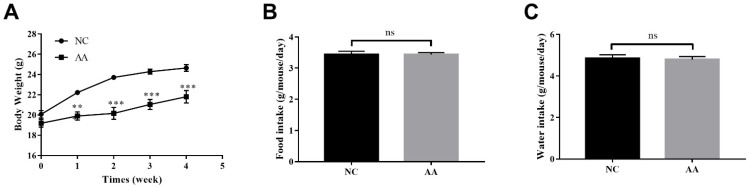
(**A**) Body weight, (**B**) food intake, and (**C**) water intake of mice fed with different diet. Results are expressed as mean ± SEM (*n* = 8). ** *p* < 0.01, *** *p* < 0.001, compared to the NC group based on ANOVA with Sidak’s multiple comparisons test. NC: normal control; AA: Acrylamide; ns, not significant.

**Figure 2 foods-10-02990-f002:**
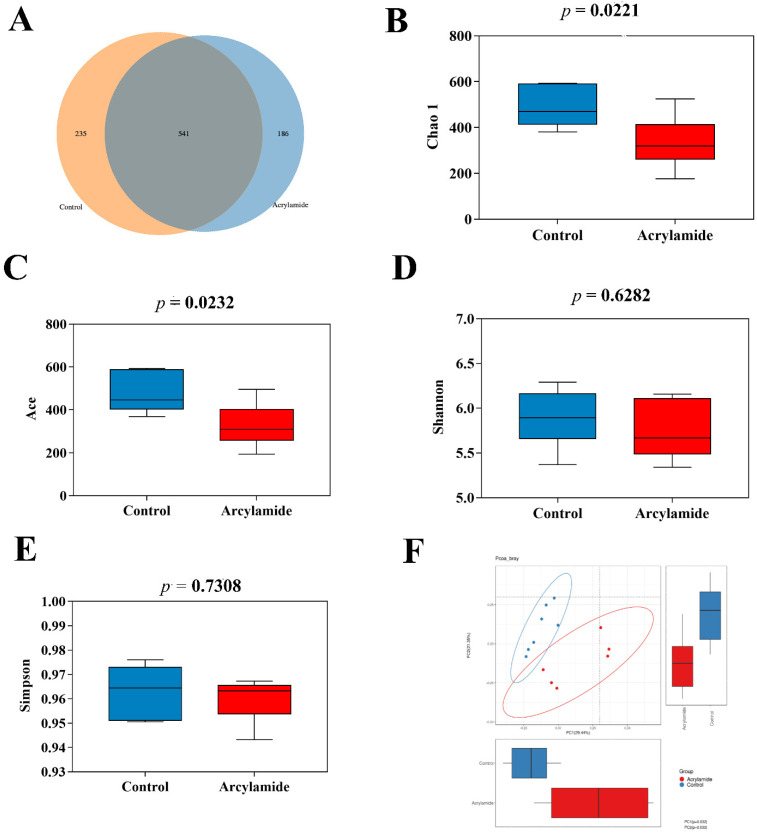
Comparison of α-diversity and β-diversity analysis of gut microbiota between NC and AA groups. (**A**) The Venn diagram of the two groups. (**B**) Chao 1 diversity index. (**C**) Ace diversity index. (**D**) Shannon diversity index. (**E**) Simpson’s diversity index. (**F**) PcoA plot analysis based on weighted UniFrac metric.

**Figure 3 foods-10-02990-f003:**
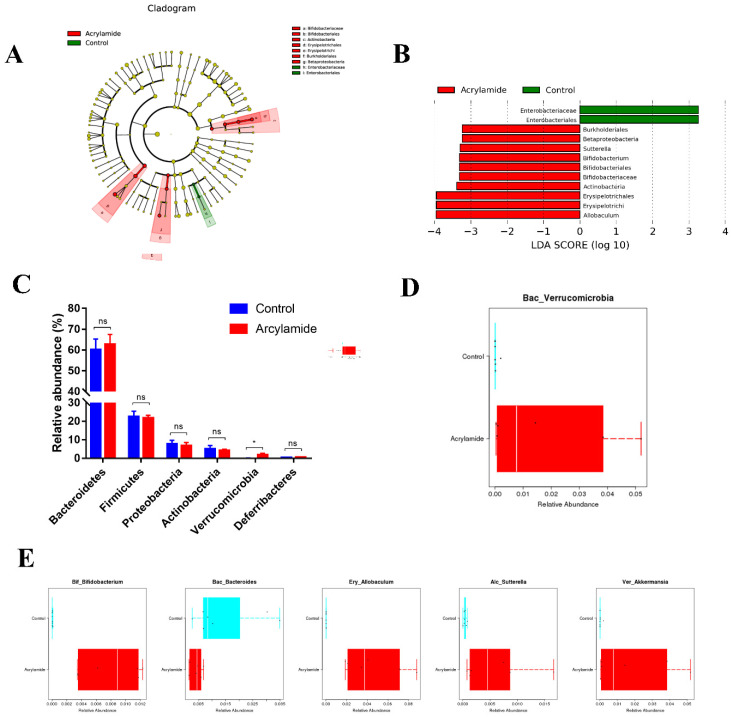
The differences of gut microbiota between NC and AA group. (**A**) Taxonomic cladogram obtained from LEfSe analysis of 16S sequences. (**B**) Taxa meeting a LDA score threshold of >2 are shown. (**C**) Comparison of fecal microbiota in control and AA groups at phylum level. (**D**) Comparison of relative abundance of significant bacteria taxa between two groups at phylum level. (**E**) Comparison of relative abundance of significant bacteria taxa between two groups at genum level. Data are presented as the means ± SEM. ns, not significant. * *p* < 0.05, compared to the control group based on ANOVA with Sidak’s multiple comparisons test.

**Figure 4 foods-10-02990-f004:**
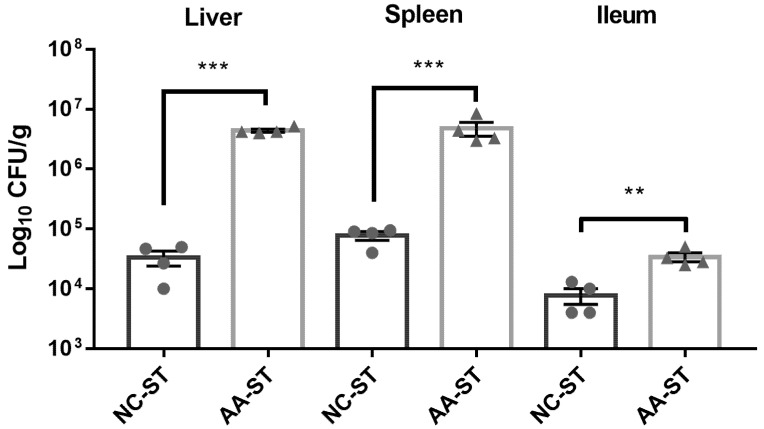
Dissemination of *S*. Typhimurium in orally infected control or AA-treated mice was detected on day 5 post-infection by measuring CFU per gram of liver, spleen, and ileum. (*n* = 4, standard deviation from the mean, statistical significance was conducted by using Mann-Whitney Nonparametric test). ** *p* < 0.01, *** *p* < 0.001. Error bars represent SEM.

**Figure 5 foods-10-02990-f005:**
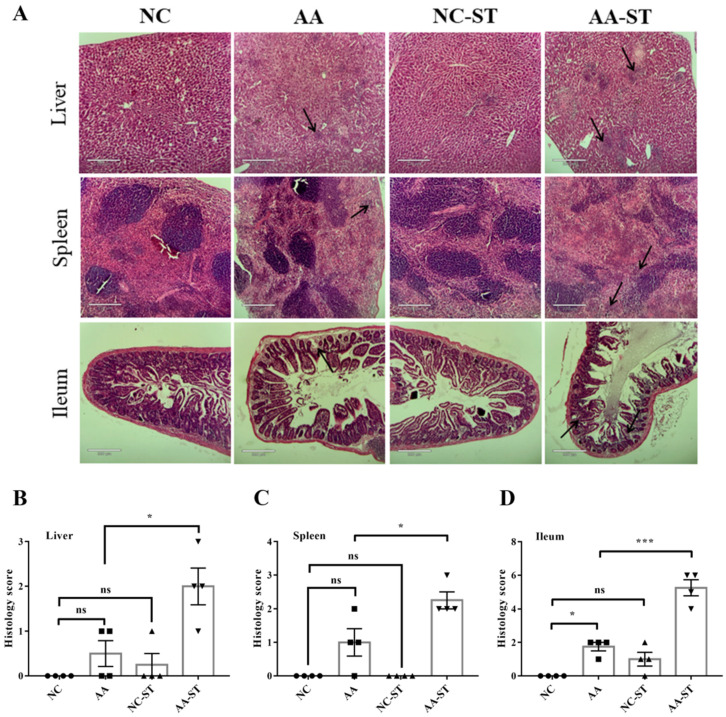
The representative light microscopic images of H&E (**A**) and histology score in liver (**B**), spleen (**C**), and ileum (**D**) on day 5 after 1 × 10^8^ CFU of *S*. Typhimurium infection. Arrows (black) indicate mild-to-moderate multifocal hepatic necrosis and splenic necrosis and inflammatory cells in the ileum of AA-treated and AA-ST challenged mice. Standard deviation from the mean, statistical significance was conducted by using one-way ANOVA with Tukey’s multiple comparisons test. * *p* < 0.05, *** *p* < 0.001, ns, not significant. Error bars represent SEM.

**Figure 6 foods-10-02990-f006:**
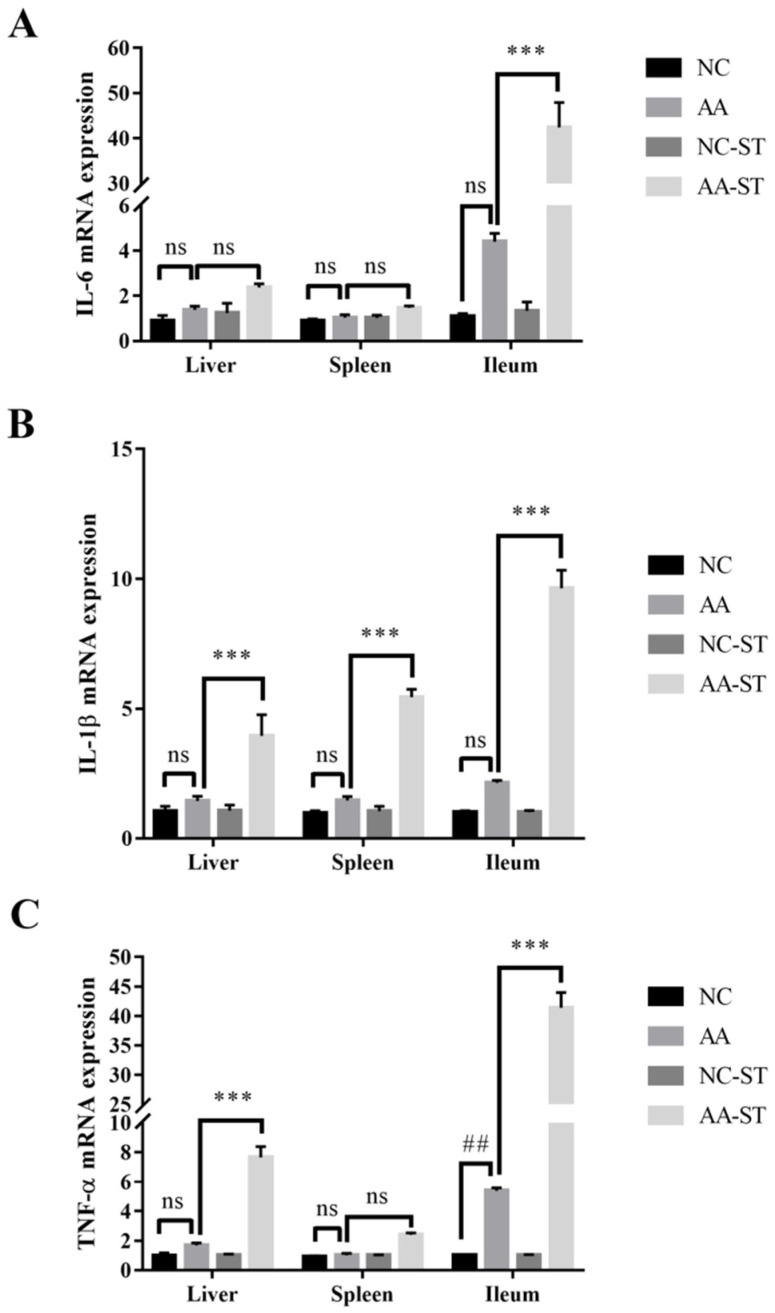
Transcriptional expression of pro-inflammatory cytokines (**A**) IL-6, (**B**) TNF-α, and (**C**) IL-1β. Standard deviation from the mean, statistical significance was conducted by using two-way ANOVA with Tukey’s multiple comparisons test. *** *p* < 0.001 for AA vs. AA-ST, ## *p* < 0.01, for NC vs. AA, ns, not significant. Error bars represent SEM.

**Figure 7 foods-10-02990-f007:**
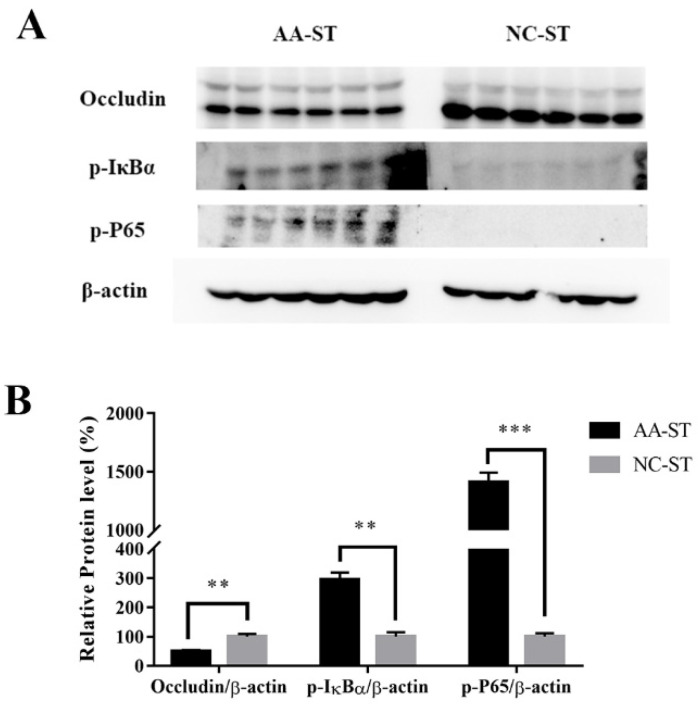
Immunoblot images analysis (**A**) and band intensity analysis (**B**) of Occludin, p-IκBα, and p-P65 from ileum of mice. Standard deviation from the mean, statistical significance was conducted by using Mann-Whitney Nonparametric test. ** *p* < 0.01, *** *p* < 0.001. Error bars represent SEM.

**Figure 8 foods-10-02990-f008:**
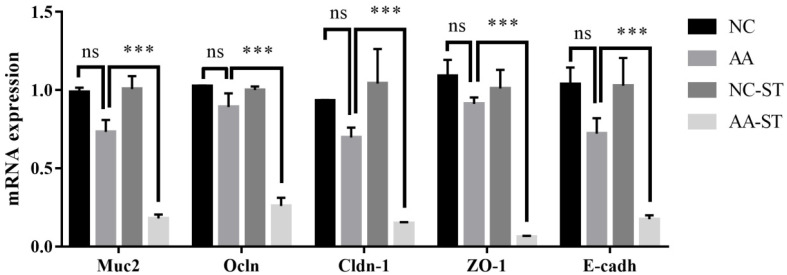
The mRNA expression level of Mucin2 (Muc2), E-cadherin (E-cadh), Occludin (Ocln), Claudin-1 (Cldn-1), and zonula-occludens (ZO)-1 in ileum. Standard deviation from the mean, statistical significance was conducted by using two-way ANOVA with Tukey’s multiple comparisons test. *** *p* < 0.001 for AA vs. AA-ST, ns, not significant. Error bars represent SEM.

## Data Availability

The datasets generated for this study are available from the authors.

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
