# Peer review of "Dietary Acrylamide Intake Alters Gut Microbiota in Mice and Increases Its Susceptibility to Salmonella Typhimurium Infection"

_foods, 2021, doi:10.3390/foods10122990_

Round 1

Reviewer 1 Report

I have read the work of Wang and co-workers on an interesting experiment of susceptibilities of mice gut microflora do the feeding with Acrylamide.

The undertaken work seems very detailed, but my main question would be that the authors have investigated the water enriched with Acrylamide and not a food, could this cause some difference in the absorption of the Acrylamide?

another question is how the authors have assessed and decided the amount of acrylamide to be administered to the mice? this should be evaluated and referenced and be similar to the human exposure to acrylamide.

do the authors have analysed the free acrylamide in the mice fluids?

Author Response

1. The undertaken work seems very detailed, but my main question would be that the authors have investigated the water enriched with Acrylamide and not a food, could this cause some difference in the absorption of the Acrylamide?

Response: Thanks for the suggestion. In this study, mice were administrated with acrylamide by oral gavage to accurately control the amount of acrylamide intake. Although previous study had indicated that there was no apparent influence of dietary fibre and fat on the uptake and absorption of acrylamide from the gastrointestinal tract [1-2], we still agree with the reviewer’s comment that some other food components may cause certain difference in the absorption and we will take this into account in our further experiment.

2. another question is how the authors have assessed and decided the amount of acrylamide to be administered to the mice? this should be evaluated and referenced and be similar to the human exposure to acrylamide.

Response: Average intake of acrylamide for the general population estimated by FAO/WHO range from 0.3 to 0.8 μg/kg body weight per day [3]. Considering different dietary patterns among various countries, we decided to use half of the lower range (0.3/2=0.15μg/kg bw) as the dose for the study, which corresponds to approximately 1 μg/kg bw for mice. Even at this low dosage, acrylamide still caused changes in gut microbiota and affected susceptibility of mice to Salmonella Typhimurium infection.

3. do the authors have analysed the free acrylamide in the mice fluids?

Response: The primary metabolite of acrylamide is glycidamide. Moreover, acrylamide and its metabolites are rapidly eliminated in the urine, primarily as mercapturic acid conjugates of acrylamide and glycidamide [4]. Therefore, we have not analysed the free acrylamide in the mice fluids in the manuscript.

References

[1] Woo, G.H., et al. Lack of preventive effects of dietary fibers or chlorophyllin against acrylamide toxicity in rats. Food and Chemical Toxicology, 2007, 45, 1507-1515.

[2] Sánchez, J., et al. Formation of hemoglobin adducts of acrylamide after its ingestion in rats is dependent on age and sex. Journal of Agricultural and Food Chemistry, 2008, 56, 5096-5101.

[3] FAO/WHO consultation on the health implications of acrylamide in food, 25-27 June 2002. Geneva, Food and Agriculture Organization of the United Nations/World Health Organization.

[4] Safety evaluation of certain contaminants in food. WHO Food Additives Series, No. 55/FAO Food and Nutrition Paper, No. 82, 2006.

Reviewer 2 Report

Please revise the spelling, I have detected a couple of typos: Line 133 there is a typo. 0=noralm; I guess it should be normal line 183, divesity,

Line 151, please indicate the meaning of LB,  and indicate the brand in Reagents

Line 152, please indicate incubation temperature for microbiological analyses

Line 191,  the sentence “Abovementioned results indicated that  AA group showed no remarkable effect  on  the  gut  microbiota  diversity  compared  to  NC  group,  but  AA  significantly  reduced gut microbiota species richness compared to NC group” seems incoherent, please rewrite

Line 231 3.3. The effect of AA on host susceptibility to S. Typhimurium infection The text in line 232-238 is the same than in lines 172-178. Besides, the is no text referring results showed in figure 4. Please rewrite

Line 296 3.4. The effect of AA on inflammatory cytokines. This section title is the same as in line 272 I suggest to rename as 3.5 the effect of AA on intestine epithelial  structure, or to delete the section title

Line 304,  please revise the sentence,  “In  addition,  the  expression  of  occludin  was remarkably  suppressed  in  AA treated mice (Figure 7) and consider to maintain the formula “AA-ST group compared to that of AA group” as in the previous sentence.

Author Response

1. Please revise the spelling, I have detected a couple of typos: Line 133 there is a typo. 0=noralm; I guess it should be normal line 183, divesity,

Response: According to reviewer’s comments, we have revised the spelling in revised manuscript as follows: 0=normal in line 135, The effect of AA on the diversity and composition of gut microbiota in line 186.

2. Line 151, please indicate the meaning of LB,  and indicate the brand in Reagents

Response: The meaning of LB is Luria-Bertani. According to reviewer’s comments, we have added the brand of LB in revised manuscript as follows: Luria-Bertani (LB) medium was purchased from Qingdao Hope Bio-Technology Co., Ltd.

3. Line 152, please indicate incubation temperature for microbiological analyses

Response: According to reviewer’s comments, the incubation temperature for microbiological analyses has been supplied in revised manuscript as follows: After incubated in 37°C for 24 h in line 154.

4. Line 191,  the sentence “Abovementioned results indicated that  AA group showed no remarkable effect  on  the  gut  microbiota  diversity  compared  to  NC  group,  but  AA  significantly  reduced gut microbiota species richness compared to NC group” seems incoherent, please rewrite

Response: According to reviewer’s comments, the sentence has been rewritten in revised manuscript as follows: Abovementioned results indicated that AA group showed no significant effect on the gut microbiota diversity compared to NC group in line 193-195.

5. Line 231 3.3. The effect of AA on host susceptibility to S. Typhimurium infection The text in line 232-238 is the same than in lines 172-178. Besides, the is no text referring results showed in figure 4. Please rewrite

Response: According to reviewer’s comments, the text has been rewritten in revised manuscript as follows: We subsequently investigated the host susceptibility to oral S. Typhimurium infection after AA consumption. As shown in figure 4, the level of S. Typhimurium in some internal organs such as liver (p < 0.001), spleen (p < 0.001), and ileum (p < 0.01) were significantly increased in mice with AA diet compared to that of NC group. Moreover, the level of S. Typhimurium in liver and spleen was significantly higher than that in ileum in both NC-ST and AA-ST group. The results indicated that feeding of AA diet for 4 weeks significantly increased host susceptibility to oral S. Typhimurium infection compared to NC group in line 233-240.

6. Line 296 3.4. The effect of AA on inflammatory cytokines. This section title is the same as in line 272 I suggest to rename as 3.5 the effect of AA on intestine epithelial  structure, or to delete the section title

Response: According to reviewer’s comments, the section title in line 298 has been renamed as 3.5 the effect of AA on intestine epithelial structure in revised manuscript.

7. Line 304,  please revise the sentence,  “In  addition,  the  expression  of  occludin  was remarkably  suppressed  in  AA treated mice (Figure 7) and consider to maintain the formula “AA-ST group compared to that of AA group” as in the previous sentence.

Response: According to reviewer’s comments, the sentence has been revised in revised manuscript as follows: In addition, the expression of occludin was remarkably suppressed in AA-ST group compared to that of NC-ST group (Figure 7) in line 306-307.